# Amelioration of Hepatic Steatosis in Mice through *Bacteroides uniformis* CBA7346-Mediated Regulation of High-Fat Diet-Induced Insulin Resistance and Lipogenesis

**DOI:** 10.3390/nu13092989

**Published:** 2021-08-27

**Authors:** Hye-Bin Lee, Moon-Ho Do, Hyunjhung Jhun, Sang-Keun Ha, Hye-Seon Song, Seong-Woon Roh, Won-Hyong Chung, Young-Do Nam, Ho-Young Park

**Affiliations:** 1Research Division of Food Functionality, Korea Food Research Institute, Wanju 55365, Korea; 50023@kfri.re.kr (H.-B.L.); Do.moon-ho@kfri.re.kr (M.-H.D.); skha@kfri.re.kr (S.-K.H.); whchung@kfri.re.kr (W.-H.C.); 2Department of Food Science and Technology, Jeonbuk National University, Wanju 54896, Korea; 3Technical Assistance Center, Korea Food Research Institute, Jeonju 55365, Korea; vetian@kfri.re.kr; 4Microbiology and Functionality Research Group, World institute of Kimchi, Gwangju 61755, Korea; ladolcevita486@gmail.com (H.-S.S.); seong18@gmail.com (S.-W.R.)

**Keywords:** *Bacteroides uniformis*, high-fat diet, non-alcoholic fatty liver disease, hepatic steatosis, lipid metabolism

## Abstract

Dietary habits and gut microbiota play an essential role in non-alcoholic fatty liver disease (NAFLD) and related factors such as insulin resistance and de novo lipogenesis. In this study, we investigated the protective effects of *Bacteroides uniformis* CBA7346, isolated from the gut of healthy Koreans, on mice with high-fat diet (HFD)-induced NAFLD. Administration of *B. uniformis* CBA7346 reduced body and liver weight gain, serum alanine aminotransferase and aspartate aminotransferase levels, liver steatosis, and liver triglyceride levels in mice on an HFD; the strain also decreased homeostatic model assessment for insulin resistance values, as well as serum cholesterol, triglyceride, lipopolysaccharide, leptin, and adiponectin levels in mice on an HFD. Moreover, *B. uniformis* CBA7346 controlled fatty liver disease by attenuating steatosis and inflammation and regulating de novo lipogenesis-related proteins in mice on an HFD. Taken together, these findings suggest that *B. uniformis* CBA7346 ameliorates HFD-induced NAFLD by reducing insulin resistance and regulating de novo lipogenesis in obese mice.

## 1. Introduction

Non-alcoholic fatty liver disease (NAFLD) is the most common chronic liver disease, characterized by irregular lipid accumulation in hepatocytes even without alcohol abuse [1]. The global prevalence of NAFLD is increasing at a rate comparable to that of type 2 diabetes mellitus and obesity, and is estimated to be present in 25% of adults [2]. As NAFLD progresses, serious liver diseases including hepatic inflammation, hepatic fibrosis, cirrhosis, liver failure, and liver cancer develop [3]. Thus, there is a great need for the discovery of efficient therapeutic agents for the prevention and treatment and of NAFLD.

The pathogenesis of NAFLD has not yet been clearly elucidated, and the main hypothesis that explains the progression of NAFLD is the “multiple-hit hypothesis”. A variety of factors including dietary habits such as a high-fat diet (HFD), and genetic and environmental factors, can lead to obesity accompanied by insulin resistance and adipocyte proliferation [4]. Insulin resistance plays a key role in the pathogenesis of steatosis, and increases hepatic de novo lipogenesis [5]. Increases in circulating insulin and glucose related to insulin resistance are reported to be involved in the stimulation of hepatic de novo lipogenesis, because glucose and insulin activate the carbohydrate response element binding protein (ChREBP) and sterol regulatory element-binding protein 1 (SREBP-1), respectively, which activate genes associated with de novo lipogenesis [6,7]. Therefore, controlling insulin resistance and inhibiting the de novo lipogenesis pathway could reduce hepatic steatosis in obese individuals.

Many recent studies have linked the pathogenesis of NAFLD to changes in the gut microbiome [8,9]. In addition, numerous studies have shown that probiotics such as *Lactobacillus* and *Bifidobacterium* can ease NAFLD symptoms in animal models [10,11]. Although current research is predominantly on commercial probiotics, it is expected that other native strains that are the dominant inhabitants of the human gut may also constitute the next generation of probiotics to be studied [12]. The *Bacteroides uniformis* strain, one of the predominant constituents of the human gut microbiota, did not cause safety concerns, and showed anti-obesity effects in animal models [12,13]. We hypothesized that an increase in the number of *B. uniformis* isolated from the intestines of healthy Koreans could help control hepatic steatosis and obesity.

However, the protective effects of *B. uniformis*—isolated from the feces of healthy humans—against HFD-induced NAFLD have not been reported. Hence, in this study, we investigated whether *B. uniformis* CBA7346 has a protective role against HFD-induced NAFLD in mice.

## 2. Materials and Methods

### 2.1. Bacterial Strain and Culture Conditions

A healthy human isolate—*B. uniformis* CBA7346—was provided by the World Institute of Kimchi. The 16S rRNA gene sequence extracted from the genome sequence of *B. uniformis* CBA7346 showed the closest similarity to *B. uniformis* ATCC 8492^T^ (99.9%). The 16S rRNA gene sequences of CBA7346 and its related strains (>99% similarity to the 16S rDNA sequence) were aligned, and a distance matrix was constructed using the MEGA X software [14]. The corresponding phylogenetic tree was reconstructed using the maximum likelihood method and Kimura 2-parameter model (bootstrap 1000 replicates) [15] (Appendix A).

*B. uniformis* was activated in brain heart infusion (BHI) agar (BD Biosciences, Franklin Lakes, NJ, USA) and BHI broth (BD Biosciences) at 37 °C under anaerobic conditions in an anaerobic jar system (Mitsubishi Gas Chemical, Tokyo, Japan) with AnaeroPack (Mitsubishi Gas Chemical). After activation in BHI broth, the strain was centrifuged (3000× *g*, 5 min) to a density of 1 × 10^7^ CFU/mL, and then suspended in phosphate-buffered saline (PBS; HyClone, Logan, UT, USA) before use.

### 2.2. In Vivo Mouse Model Study

#### 2.2.1. Mice and Diets

Seven-week-old male C57BL/6J mice were procured from the Central Laboratory Animal (Seoul, Korea), and all experiments were approved by the Korea Food Research Institutional Animal Care and Use Committee (KFRI-M-20013). The mice were kept in an environment maintained at 22 ± 2 °C and 55 ± 5% humidity with a 12 h light/dark cycle, and fed an Ain-93G diet (Dyets, Bethlehem, PA, USA) and sterilized water.

After the adaptation period (7 days), the mice were randomly divided into four groups (*n* = 9 per group) as follows: (1) normal control group, receiving an Ain-93G diet and 100 μL of PBS by oral administration (NC) thrice a week; (2) a group receiving an Ain-93G diet and a dose of 1 × 10^6^ CFU/100 μL *B. uniformis* CBA7346 by oral administration (NU) thrice a week; (3) an obese group, receiving a 60% kcal fat diet (HC; TD.06414, Harlan, Madison, WI, USA) and 100 μL of PBS by oral administration thrice a week; and (4) a group receiving a 60% kcal fat diet and a dose of 1 × 10^6^ CFU/100 μL *B. uniformis* CBA7346 by oral administration (HU) thrice a week. Body weight and food intake were measured once a week during the 12-week study period. After 12 weeks, the mice were made to fast overnight, and euthanized by isoflurane anesthesia. Blood was collected from the abdominal vena cava and centrifuged at 3000× *g* for 15 min. Liver tissues were removed and stored at −70 °C.

#### 2.2.2. Blood Serum Analysis

The alanine aminotransferase (ALT), aspartate aminotransferase (AST), total cholesterol, low-density lipoprotein (LDL) cholesterol, and high-density lipoprotein (HDL) cholesterol levels were measured using a commercial kit from Cell Biolabs (Beverly, MA, USA), triglyceride levels were measured using the Triglyceride Colorimetric Assay Kit (Cayman Chemical, Ann Arbor, MI, USA), endotoxin levels were measured using the Pierce^TM^ LAL Chromogenic Endotoxin Quantitation Kit (Thermo Fisher Scientific, Waltham, MA, USA), and leptin and adiponectin levels were analyzed using a commercial kit from R&D Systems (Minneapolis, MN, USA). All experiments were performed according to the manufacturer’s instructions.

#### 2.2.3. Homeostatic Model Assessment for Insulin Resistance (HOMA-IR)

HOMA-IR was performed using insulin and glucose concentrations obtained after 12 h of fasting at 12 weeks, using the following equation [16]:(1)HOMA−IR ={fasting insulin (U/L)×fasting glucose (mg/dL)}/405

#### 2.2.4. Western Blotting

The mice livers were lysed in PRO-PREP^TM^ (iNtRON Biotechnology, Seongnam, Korea), and the protein concentrations were measured using a protein assay kit from Bio-Rad (Hercules, CA, USA). Proteins were separated by SDS-PAGE, and the blots were transferred to polyvinylidene difluoride membranes. The membranes were incubated overnight at 4 °C with primary antibodies against SREBP-1, fatty acid synthase (FAS), phospho-AMP-activated protein kinase alpha (p-AMPKα), acetyl-CoA carboxylase (ACC), peroxisome proliferator-activated receptor gamma (PPARγ), and mammalian target of rapamycin (mTOR) purchased from Abcam (Cambridge, MA, USA), and ChREBP, stearoyl-CoA desaturase-1 (SCD1), AMPKα, and β-actin purchased from Cell Signaling Technology (Beverly, MA, USA). The membranes were incubated at room temperature with secondary antibodies, and bands were visualized using a ChemiDoc XRS + imaging system (Bio-Rad).

#### 2.2.5. Hepatic Histological Analysis

For histological analysis, hematoxylin and eosin (H&E) staining, and oil red O (ORO) staining were performed. Liver tissues were fixed in 10% formalin solution (Sigma-Aldrich, St. Louis, MO, USA), embedded in paraffin, sectioned (4 μm), and stained with H&E for evaluation of liver injury. The results of H&E staining were used to calculate the NAFLD activity score (NAS), considering steatosis, lobular inflammation, and hepatocellular ballooning (Table 1) [17].

Liver tissues were frozen with optimum cutting temperature compound (Sakura Finetek, Torrance, CA, USA), sectioned (4 μm), and stained with ORO for evaluation of lipid accumulation in the liver. All slide samples were scanned using a Pannoramic 250 Flash III slide scanner (3DHISTECH Ltd., Budapest, Hungary). The ORO-stained area was calculated using ImageJ software (NIH, Bethesda, MD, USA).

#### 2.2.6. Microbial Bioinformatics Analysis

Fecal samples were collected on the non-administration day at 12 weeks, and the microbiota was analyzed by sequencing the V3–V4 region of 16S rRNA using a MiSeq system (Illumina, San Diego, CA, USA) at Macrogen (Seoul, Korea), as per the manufacturer’s instructions. Pairs of reads from the DNA fragments were assembled using FLASH [18], operational taxonomic units (OTUs) were defined at sequence homology ≥ 97% [19], and taxonomy was performed using QIIME-UCLUST [20]. The QIIME software was used to measure microbial diversity based on unweighted UniFrac distance matrices [19].

### 2.3. Statistical Analysis

To assess the relative significance among the groups, Duncan’s multiple range test was performed at *p* < 0.05, and all data were expressed as the mean ± standard error of the mean. Statistical analysis was performed using IBM SPSS Statistics for Windows, version 20 (IBM Corp., Armonk, NY, USA).

## 3. Results

### 3.1. Effects of B. uniformis CBA7346 on Body Weight and Liver Weight

The body and liver weights of the HC group increased significantly (*p* < 0.05) compared to the NC group, at 12 weeks (Figure 1A,B). While *B. uniformis* CBA7346 did not affect the body and liver weights of mice on a normal diet, the HU group had significantly lower body and liver weights (*p* < 0.05) compared to the HC group at the end of 12 weeks. Moreover, the HU group exhibited a significantly lower (*p* < 0.05) liver/body weight ratio compared to the HC group (Figure 1D). However, there was no significant difference in food intake between the HC group and the HU group (Figure 1C).

### 3.2. Effects of B. uniformis CBA7346 on Metabolic Parameters

Serum insulin levels were significantly higher in the HC group than in the NC group (*p* < 0.05) (Figure 2A). In addition, the HOMA-IR value, an index of insulin resistance, was significantly higher in the HC group than in the NC group (*p* < 0.05) (Figure 2B). *B. uniformis* CBA7346 did not affect insulin and HOMA-IR levels in mice on a normal diet, but the HU group exhibited significantly lower levels (*p* < 0.05) than the HC group.

The HC group showed significantly increased levels of ALT, AST, and endotoxin (*p* < 0.05), whereas the HU group showed remarkably reduced levels (Figure 2).

### 3.3. Effects of B. uniformis CBA7346 on Serum Lipids and Hormones

HFD-induced changes in biochemical and hormonal parameters were evaluated using serum cholesterol, triglyceride, leptin, and adiponectin levels (Figure 3). No significant differences in serum lipids and hormones were observed between the NC and NU groups; *B. uniformis* CBA7346 did not affect serum lipid and hormone levels in mice on a normal diet. In contrast, the HU group showed significantly decreased levels of serum lipids and hormones (*p* < 0.05) compared to the HC group. These results suggest that *B. uniformis* CBA7346 alters HFD-induced changes in biochemical and hormonal parameters.

### 3.4. Effects of B. uniformis CBA7346 on Liver Injury and Steatosis

Histological analysis of liver tissues showed that administration of *B. uniformis* CBA7346 reduced HFD-induced steatosis and hepatocyte ballooning (Figure 4A). The HU group showed significantly lower elevations in NAS values (*p* < 0.05) compared to the HC group (Figure 4C).

As shown in Figure 4B,D, the ORO-stained area in liver tissues was significantly lower in the HU group (*p* < 0.05) than in the HC group. Furthermore, the HU group showed significantly less reduction in hepatic triglyceride levels (*p* < 0.05) compared to the HC group (Figure 4E). These results suggest that administration of *B. uniformis* CBA7346 heals HFD-induced fatty liver.

### 3.5. Effects of B. uniformis CBA7346 on Hepatic Lipid Metabolism

To evaluate whether the administration of *B. uniformis* CBA7346 can alter lipid metabolism in liver tissues, SREBP-1, ChREBP, mTOR, p-AMPK, AMPK, ACC, FAS, SCD1, and PPARγ protein expressions were measured (Figure 5). *B. uniformis* CBA7346 did not affect protein expression related to lipid metabolism in mice on a normal diet. In contrast, the HU group showed significantly decreased expression of lipid metabolism-related proteins (*p* < 0.05) compared to the HC group (except for SCD1 protein expression). In addition, the p-AMPKα/AMPKα ratio was significantly higher in the HU group than in the HC group (*p* < 0.05). These results suggest that *B. uniformis* CBA7346 alters HFD-induced changes in hepatic lipid metabolism.

### 3.6. Effects of B. uniformis CBA7346 on Gut Microbial Composition

We investigated the effects of *B. uniformis* CBA7346 on the intestinal microbiota of mice on an HFD (Figure 6). Bacteroidetes, Firmicutes, and Proteobacteria were the dominant bacteria at the phylum level (Figure 6A). We observed fewer OTUs in the HC and HU groups than in the groups on a normal diet (Figure 6B). The HU group exhibited a decrease in the Firmicute to Bacteroidete ratio (F/B) compared to the HC group, but this decrease was not significant (Figure 6C).

## 4. Discussion

HFD intake is associated with inflammation, obesity, metabolic disorders, gut dysbiosis, and NAFLD [21]. NAFLD—one of the most prevalent liver diseases in adults—is characterized by an irregular accumulation of triglycerides in hepatocytes [1,22]. Recent studies have reported the administration of probiotics such as *Lactobacillus* for the treatment and prevention of NAFLD in animal models [9,23]. *B. uniformis* which is dominant in the human gut, showed anti-obesity effects in mice with HFD-induced obesity [13]. In addition, *B. uniformis* isolated from the feces of healthy infants has the potential to be a probiotic strain, and its administration does not raise safety issues in mice [12]. The effects of *B. uniformis* CBA7346 isolated from the feces of healthy Koreans on mice with HFD-induced NAFLD have not been reported. Therefore, we investigated the anti-obesity effects of *B. uniformis* CBA7346 as well as its effects on NAFLD-related biomarkers; the gut microbiome of mice with HFD-induced NAFLD was also explored.

HFDs have been associated with lipid accumulation, insulin resistance, and metabolic diseases [1]; to induce NAFLD in a mouse model, mice were fed a 60% kcal fat diet for 12 weeks. In our mouse model, administration of *B. uniformis* CBA7346 inhibited body and liver weight gain in mice on an HFD, but had no effect on the body and liver weight of mice on a normal diet. In addition, serum insulin and HOMA-IR levels were lower in the HU group than in the HC group. An alternative to glucose clamp, HOMA-IR is the most commonly used surrogate measure of the insulin resistance in animal experiments [16]. Insulin resistance remains at the top of the NAFLD mechanism [1], and serum insulin levels are associated with lobular inflammation and hepatocyte ballooning [24]. ALT and AST, the major biomarkers used for detecting hepatic injury in NAFLD [3], were also inhibited in the HU group on an HFD. These results showed that *B. uniformis* CBA7346 administration has a protective effect against liver injury. NAFLD promotes liver inflammation; NAFLD patients therefore have disrupted gut barrier integrity and bacterial overgrowth, which can result in the release of lipopolysaccharide (LPS) through the gut-liver axis, leading to hepatocyte death and lipid accumulation [25]. Mice on an HFD, when treated with *B. uniformis* CBA7346, showed decreased LPS release in the serum confirming that *B. uniformis* CBA7346 also prevented the passage of LPS into the blood.

Hepatic steatosis plays an essential role in the progression of NAFLD, and a prolonged HFD sequentially leads to steatosis and steatohepatitis, progressing to bridging fibrosis [26,27]. Our histological analysis indicated that the administration of *B. uniformis* CBA7346 suppressed extensive hepatic steatosis, lobular inflammation, and hepatocellular ballooning in mice on an HFD. In addition, *B. uniformis* CBA7346 administration while on an HFD affected the de novo lipogenesis pathway in the liver. SREBP1 activated by insulin, and ChREBP activated by glucose regulates hepatic lipogenesis [6]. SREBP1 and ChREBP are central regulators of most hepatic lipid synthesis-related genes, including the ACC, FAS, and SCD1 [25,28]. In the HU group which is on an HFD, the expression of hepatic lipogenesis-related proteins such as the SREBP1, ChREBP, ACC, and FAS was significantly downregulated, but not that of SCD1. PPARs are known to play an important role in hepatic lipid metabolism, and PPARγ induces the production of lipid droplets upon overexpression in hepatocytes [29]. As expected, PPARγ expression was increased in the HC group, and *B. uniformis* CBA7346 efficiently attenuated the accumulation of triglycerides by downregulating this protein. Furthermore, mice on an HFD, when treated with *B. uniformis* CBA7346, showed increased AMPKα activation levels. AMPKα regulates hepatic lipid metabolism by modulating transcription factors such as SREBP-1 and ChREBP [30]. Likewise, the expression of mTOR, which plays a key role in promoting de novo lipogenesis by regulating several lipogenic genes [31], was significantly downregulated in the HU group during the period the mice were on an HFD. Therefore, our results demonstrate that *B. uniformis* CBA7346 can inhibit hepatic lipid synthesis by regulating lipid metabolism-related proteins in obese mice.

Dyslipidemias, i.e., abnormal levels of lipid profiles such as total cholesterol, LDL cholesterol, HDL cholesterol, and triglycerides, are also induced by HFDs in animals [32]. In our study, the administration of *B. uniformis* CBA7346 reduced the serum levels of lipid profiles in mice with HFD-induced NAFLD. Leptin and adiponectin—important hormones in obesity management—were also suppressed in *B. uniformis* CBA7346 treated mice on an HFD. These factors could also possibly regulate the expression of lipid homeostasis-related proteins in the liver [13].

Many studies have shown that the gut-liver axis plays an important role in NAFLD development [1,9]; we found that groups on an HFD had fewer OTUs than groups on normal diets. However, the administration of *B. uniformis* CBA7346 did not significantly influence microbial composition changes such as the changes in the F/B ratios caused by the HFD. Another study reported that daily administration of 5 × 10^8^ CFU *B. uniformis* ameliorated intestinal dysbiosis in mice with HFD-induced NAFLD [13], but we administered 1 × 10^6^ CFU *B. uniformis* CBA7346 only thrice a week. Therefore, further studies are needed to identify the dose of *B. uniformis* CBA7346 that produces the most efficient results in mice with HFD-induced NAFLD with regard to NAFLD attenuating effects.

This study demonstrates, for the first time, the effects of administrating *B. uniformis* CBA7346 isolated from the gut of healthy Koreans, on mice with HFD-induced NAFLD. Administration of *B. uniformis* CBA7346 inhibited body and liver weight gain, reduced insulin resistance, and influenced serum metabolic biomarkers related to hepatic injury and steatosis in mice on an HFD. The administration of *B. uniformis* CBA734 also healed NAFLD by attenuating steatosis, inflammation, and ballooning, and regulating de novo lipogenesis-related proteins induced by an HFD in mouse liver tissues. We conclude that *B. uniformis* CBA7346 may offer protection against HFD-induced NAFLD by regulating the release of LPS and lipid metabolism-related proteins and controlling insulin resistance (Figure 7). However, further studies are needed to elucidate their mechanisms of action, and to investigate the safety issues involved in the use of the *B. uniformis* strain used in this animal study.

## Figures and Tables

**Figure 1 nutrients-13-02989-f001:**
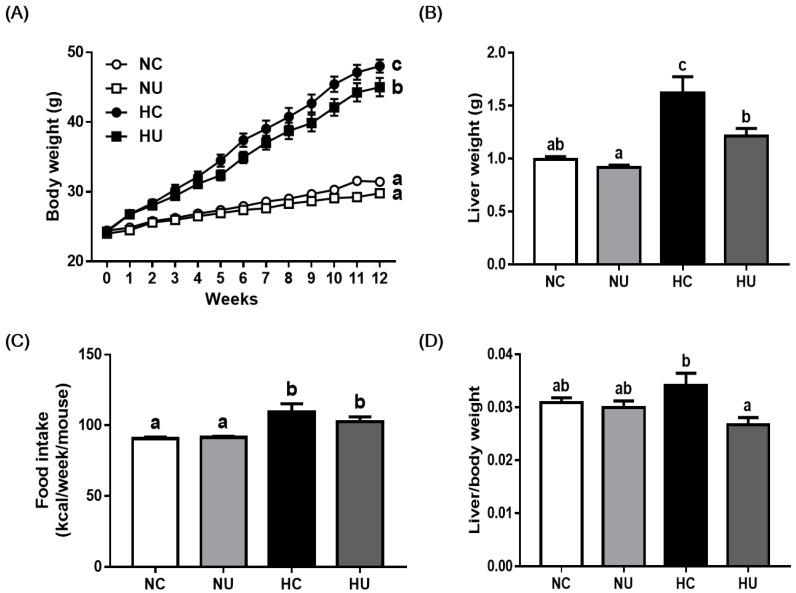
Effects of *Bacteroides uniformis* CBA7346 treatment on body and liver weight in mice with high-fat diet-induced non-alcoholic fatty liver disease. (**A**) Body weight changes during 12 weeks; (**B**) liver tissue weight (g); (**C**) food intake during the 12 weeks of feeding; (**D**) ratio of liver weight to body weight. NC, control group on normal diet; NU, group on normal diet receiving 1 × 10^6^ CFU *B. uniformis* CBA7346 thrice a week; HC, control group on high-fat diet; HU, group on high-fat diet receiving 1 × 10^6^ CFU *B. uniformis* CBA7346 thrice a week. Data are presented as mean ± SEM (*n* = 9). Means with different letters (a–c) indicate significant difference (*p* < 0.05) among groups at 12 weeks. Groups were compared using Duncan’s multiple range tests.

**Figure 2 nutrients-13-02989-f002:**
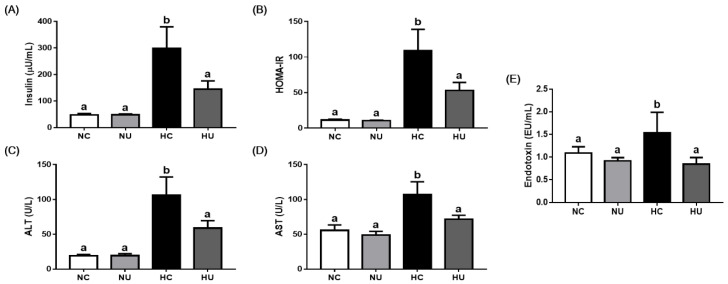
Effects of *Bacteroides uniformis* CBA7346 treatment on metabolic parameters in mice with high-fat diet-induced non-alcoholic fatty liver disease. (**A**) Serum insulin; (**B**) HOMA-IR index; (**C**) serum ALT; (**D**) serum AST; (**E**) serum LPS. NC, control group on normal diet; NU, group on normal diet receiving 1 × 10^6^ CFU *B. uniformis* CBA7346 thrice a week; HC, control group on high-fat diet; HU, group on high-fat diet receiving 1 × 10^6^ CFU *B. uniformis* CBA7346 thrice a week. Data are presented as mean ± SEM (*n* = 9). Means with different letters (a,b) indicate significant difference (*p* < 0.05) among groups. Groups were compared using Duncan’s multiple range tests.

**Figure 3 nutrients-13-02989-f003:**
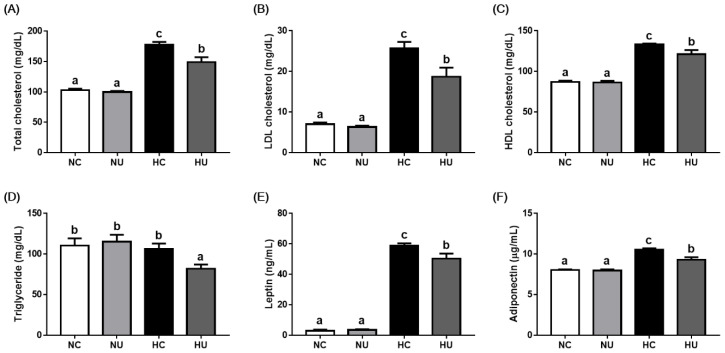
Effects of *Bacteroides uniformis* CBA7346 treatment on serum lipids and hormones in mice with high-fat diet-induced non-alcoholic fatty liver disease. (**A**) Serum total cholesterol; (**B**) serum LDL cholesterol; (**C**) serum HDL cholesterol; (**D**) serum triglyceride; (**E**) serum leptin; (**F**) serum adiponectin. NC, control group on normal diet; NU, group on normal diet receiving 1 × 10^6^ CFU *B. uniformis* CBA7346 thrice a week; HC, control group on high-fat diet; HU, group on high-fat diet receiving 1 × 10^6^ CFU *B. uniformis* CBA7346 thrice a week. Data are presented as mean ± SEM (*n* = 9). Means with different letters (a–c) indicate significant difference (*p* < 0.05) among groups. Groups were compared using Duncan’s multiple range tests.

**Figure 4 nutrients-13-02989-f004:**
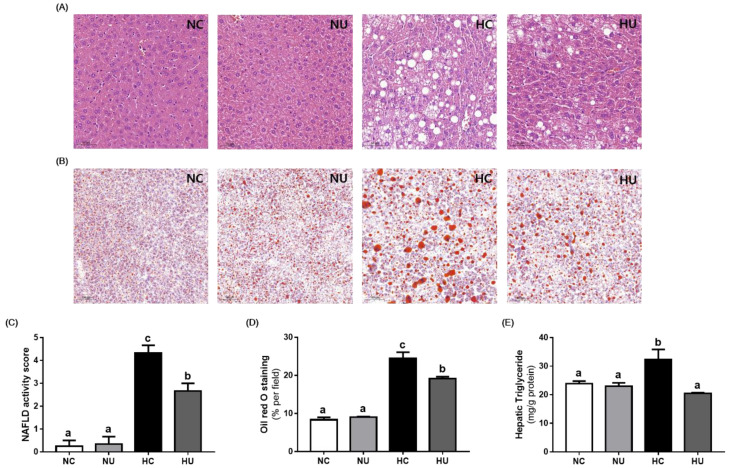
Effects of *Bacteroides uniformis* CBA7346 treatment on histological parameters of liver tissues in mice with high-fat diet-induced non-alcoholic fatty liver disease. (**A**) Representative histological results of H&E staining and (**B**) ORO staining; (**C**) quantification of NAFLD activity score and (**D**) ORO stained area; (**E**) hepatic triglyceride. NC, control group on normal diet; NU, group on normal diet receiving 1 × 10^6^ CFU *B. uniformis* CBA7346 thrice a week; HC, control group on high-fat diet; HU, group on high-fat diet receiving 1 × 10^6^ CFU *B. uniformis* CBA7346 thrice a week. Data are presented as mean ± SEM (*n* = 3). Means with different letters (a–c) indicate significant difference (*p* < 0.05) among groups. Groups were compared using Duncan’s multiple range tests.

**Figure 5 nutrients-13-02989-f005:**
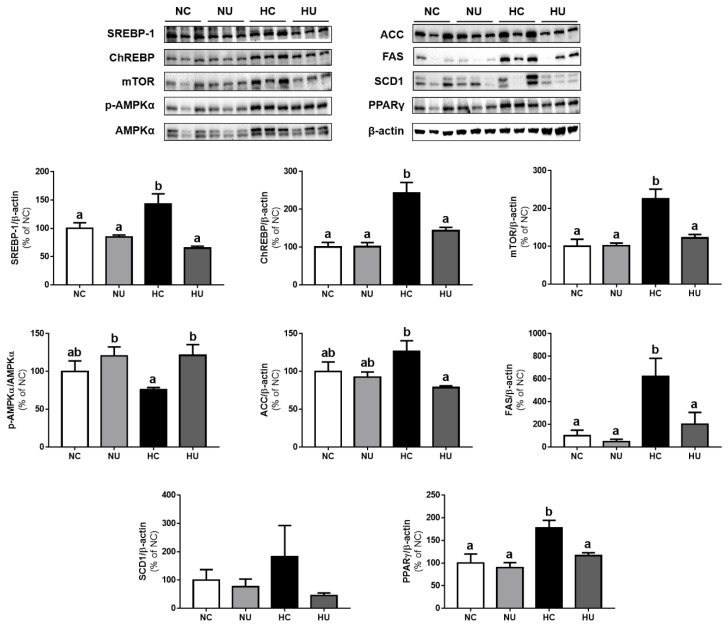
Effects of *Bacteroides uniformis* CBA7346 treatment on expression of lipid metabolism related proteins in liver tissues of mice with high-fat diet-induced non-alcoholic fatty liver disease; Western blot analysis of SREBP-1, ChREBP, mTOR, p-AMPKα, AMPKα, ACC, FAS, SCD1, and PPARγ in the liver. NC, control group on normal diet; NU, group on normal diet receiving 1 × 10^6^ CFU *B. uniformis* CBA7346 thrice a week; HC, control group on high-fat diet; HU, group on high-fat diet receiving 1 × 10^6^ CFU *B. uniformis* CBA7346 thrice a week. Data are presented as mean ± SEM (*n* = 3). Means with different letters (a,b) indicate significant difference (*p* < 0.05). Groups were compared using Duncan’s multiple range tests.

**Figure 6 nutrients-13-02989-f006:**
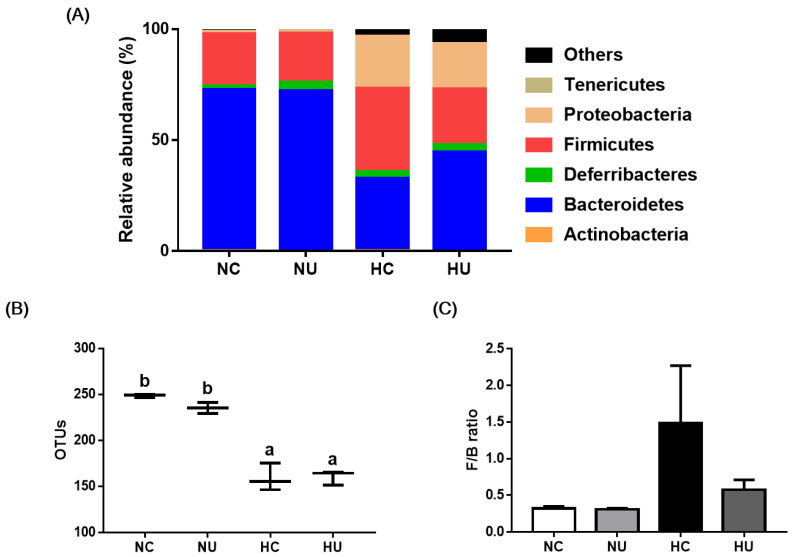
Effects of *Bacteroides uniformis* CBA7346 treatment on gut microbiota composition obtained through 16S rRNA pyrosequencing of microbiota from feces of mice with high-fat diet-induced non-alcoholic fatty liver disease. (**A**) relative abundance plot of bacteria at the phylum level; (**B**) plots of operational taxonomic units; (**C**) ratio of Firmicutes to Bacteroidetes. NC, control group on normal diet; NU, group on normal diet receiving 1 × 10^6^ CFU *B. uniformis* CBA7346 thrice a week; HC, control group on high-fat diet; HU, group on high-fat diet receiving 1 × 10^6^ CFU *B. uniformis* CBA7346 thrice a week. Data are presented as mean ± SEM (*n* = 3). Means with different letters (a,b) are significantly different (*p* < 0.05). Groups were compared using Duncan’s multiple range tests.

**Figure 7 nutrients-13-02989-f007:**
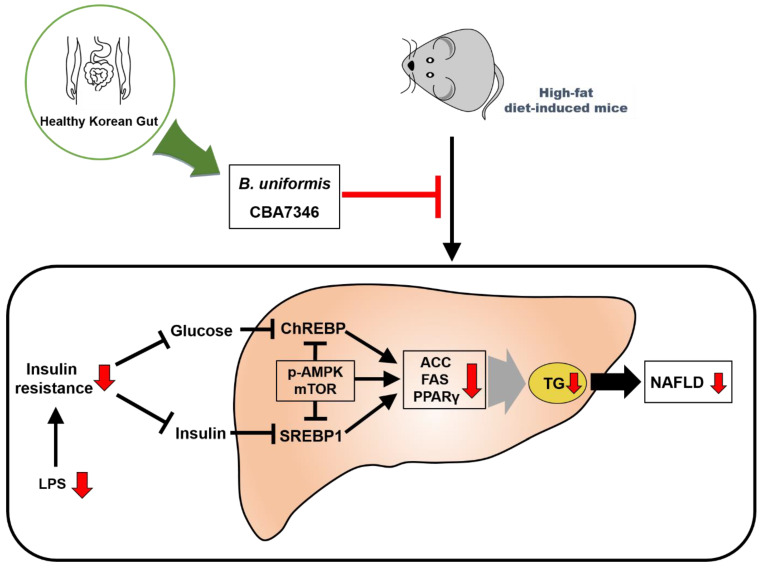
Presumed mechanism for how *Bacteroides uniformis* CBA7346 interfered with lipid accumulation in mice with high-fat diet-induced non-alcoholic fatty liver disease. LPS: lipopolysaccharide. ChREBP: carbohydrate response element binding protein. SREBP1: sterol regulatory element-binding protein 1. p-AMPK: phospho-AMP-activated protein kinase. mTOR: mammalian target of rapamycin. ACC: acetyl-CoA carboxylase. FAS: fatty acid synthase. PPARγ: peroxisome proliferator-activated receptor gamma. TG: Triglyceride. NAFLD: non-alcoholic fatty liver disease.

**Table 1 nutrients-13-02989-t001:** NAFLD activity score (NAS) system.

Item	Score	Extent
Steatosis	0	<5%
	1	5–33%
	2	34–66%
	3	>66%
Lobular Inflammation	0	No foci
	1	<2 foci per 200× field
	2	2–4 foci per 200× field
	3	>4 foci per 200× field
Hepatocyte Ballooning	0	None
	1	Few balloon cells
	2	Many cells/prominent ballooning

## Data Availability

Not applicable.

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
