# Peer review of "Amelioration of Hepatic Steatosis in Mice through Bacteroides uniformis CBA7346-Mediated Regulation of High-Fat Diet-Induced Insulin Resistance and Lipogenesis"

_nutrients, 2021, doi:10.3390/nu13092989_

Round 1
Reviewer 1 Report
My main concern is that this study lack novelty because there are several studies indicating the beneficial effects of probiotics on Fatty Liver in human. I think that showing these effects in experimental models is not new; however, the writing and presentation of the manuscript is good.
Author Response
Reviewer #1:
My main concern is that this study lack novelty because there are several studies indicating the beneficial effects of probiotics on Fatty Liver in human. I think that showing these effects in experimental models is not new; however, the writing and presentation of the manuscript is good.
â–¶ Thanks for the reviewer’s kind comments. Since many studies on development of probiotics such as Lactobacillus have been conducted, we tried to find novelty by revealing NAFLD preventive effects of intestinal strain isolated healthy Koreans. In the further study, we will proceed with the study by supplementing it to have novelty.

Reviewer 2 Report
The manuscript entitled “Amelioration of hepatic steatosis in mice through Bacteroides uniformis CBA7346-mediated regulation of high-fat diet-induced insulin resistance and lipogenesis” submitted by Hye-Bin Lee et al. for possible publication in Nutrients, describes the role of Bacteroides Uniformis CBA7346 on hepatic steatosis in mice.
I feel the manuscript needs revision.
It is not clear how many mice were studied in the main experiments (n=9 is for all experiments?). The authors should include N for each experiment in the Methods and in each Figure legend.
In methods, authors describe how mice are treated with CBA7346 but they did not clearly indicate in which volume the bacteria are delivered (i.e. the volume of gavage). If I well understand the Bacterial preparation, I think that it is 0.1ml for the 10^6 CHU per mice. Is-it this? In this way, did control mice were gavage with PBS? It must be mentioned. More other, is it really oral force-feeding or intra-gastric gavage? Regardless of the method used, the food intake of the mice should be measured. Indeed, all the effects could simply be explained by a lower food intake of the treated animals. Authors should provide this information. Food intake should be measured throughout the diet.
In the study treatment begins together with the diet. It would be interesting to test the effect of bacteria after the 12 weeks of diet in order to see if the effect exists in this situation. In other words, is it a future treatment or bacteria for NAFLD prevention?
The authors did not test and discuss the effect of the deactivated (killed) bacteria. It would be interesting to have a group of animals treated with the bacteria killed.
Do the authors find the specific bacteria in feces by sequencing (or PCR) after the treatment?
In Fig. 5 protein expression can be more discussed. Visually it looks like that B-actin is not the same in all groups suggesting that depos are not equal between sample. Could authors try to analyze another control protein such as tubulin or show red ponceau coloration?
It is important because all changes between groups depends on this. In fact, how the author explains the change in AMPKalpha (total)? It is normally stable.
Most of the findings suggest that effects of CBA 7346 is linked to the gut-liver axis and then to LPS release in the serum. What is about the histology of the intestine? Authors should give information about this. For example, with goblet cells labeling.
I have a concern about results of glycemia-insulinemia-HOMA-IR. The authors state that the treatment improves insulin resistance. It is not specifically measured (only HOMA). The blood glucose and insulinemia values ​​must be detailed.
An ITT metabolic test should be performed. It could also be supplemented by a specific pyruvate tolerance test for hepatic metabolism.
Finally, a good way to estimate liver tissue insulin resistance is to do phospho-PKB blots after insulin stimulation. It could be done and interesting things could be observed.
Despite these comments, the work is interesting.
Author Response
Reviewer #2:
The manuscript entitled “Amelioration of hepatic steatosis in mice through Bacteroides uniformis CBA7346-mediated regulation of high-fat diet-induced insulin resistance and lipogenesis” submitted by Hye-Bin Lee et al. for possible publication in Nutrients, describes the role of Bacteroides Uniformis CBA7346 on hepatic steatosis in mice.
I feel the manuscript needs revision.
It is not clear how many mice were studied in the main experiments (n=9 is for all experiments?). The authors should include N for each experiment in the Methods and in each Figure legend.
â–¶ In response to the reviewer’s comment, we revised the manuscript in each figure legend. Nine mice per group were studied in the Figure 1–3, and three mice per group were studied in the Figure 4–6.
Line 171: Data are presented as mean ± SEM (n = 9).
Lines 186–187: Data are presented as mean ± SEM (n = 9).
Line 202: Data are presented as mean ± SEM (n = 9).
Line 219: Data are presented as mean ± SEM (n = 3).
Line 237: Data are presented as mean ± SEM (n = 3).
Line 251: Data are presented as mean ± SEM (n = 3).
In methods, authors describe how mice are treated with CBA7346 but they did not clearly indicate in which volume the bacteria are delivered (i.e. the volume of gavage). If I well understand the Bacterial preparation, I think that it is 0.1ml for the 10^6 CHU per mice. Is-it this? In this way, did control mice were gavage with PBS? It must be mentioned. More other, is it really oral force-feeding or intra-gastric gavage? Regardless of the method used, the food intake of the mice should be measured. Indeed, all the effects could simply be explained by a lower food intake of the treated animals. Authors should provide this information. Food intake should be measured throughout the diet.
â–¶ Thank for the reviewer’s critical point. In 80–82 lines, we wrote that the strain suspended in PBS to a density of 1 × 107 CFU/mL. According to the reviewer’s comment, we revised the manuscript in method as below:
Lines 92–98: (1) normal control group, receiving an Ain-93G diet and 100 μL of PBS by oral administration (NC) thrice a week; (2) a group receiving an Ain-93G diet and a dose of 1 × 106 CFU/100 μL B. uniformis CBA7346 by oral administration (NU) thrice a week; (3) an obese group, receiving a 60 % kcal fat diet (HC; TD.06414, Harlan, Madison, WI, USA) and 100 μL of PBS by oral administration thrice a week; and (4) a group receiving a 60 % kcal fat diet and a dose of 1 × 106 CFU/100 μL B. uniformis CBA7346 by oral administration (HU) thrice a week.
In addition, in response to the reviewer’s comment, we inserted food intake results in the manuscript.
Lines 98–99: Body weight and food intake were measured once a week during the 12-week study period.
Lines 164–165: However, there was no significant difference in food intake between the HC group and the HU group (Figure 1 (C)).
Lines 168–169: (C) food intake during the 12 weeks of feeding; (D) ratio of liver weight to body weight.
In the study treatment begins together with the diet. It would be interesting to test the effect of bacteria after the 12 weeks of diet in order to see if the effect exists in this situation. In other words, is it a future treatment or bacteria for NAFLD prevention?
â–¶ We administered B. uniformis along with the high-fat diet for 12 weeks for the purpose of preventing NAFLD, not treatment. Referring to the reviewer’s opinion, we plan to check the potential as a therapeutic agent by adjusting the dosage of B. uniformis in the future.
The authors did not test and discuss the effect of the deactivated (killed) bacteria. It would be interesting to have a group of animals treated with the bacteria killed.
â–¶ The main purpose of this study is to evaluate the probiotic efficacy of B. uniformis. Since this is the first study of B. uniformis isolated from healthy Koreans, we decided it would be important to first understand the efficacy of live bacteria. In addition, FAO/WHO Working Group defined probiotics as “live micro-organisms which when administered in adequate amounts confer a health benefit on the host”
Do the authors find the specific bacteria in feces by sequencing (or PCR) after the treatment?
â–¶ B. uniformis was administered thrice a week, and on 12 weeks, fecal samples were collected on day when B. uniformis was not administered. According to the reviewer’s comment, we revised the manuscript in method as below:
Line 144: Fecal samples were collected on the non-administration day at 12 weeks
In Fig. 5 protein expression can be more discussed. Visually it looks like that B-actin is not the same in all groups suggesting that depos are not equal between sample. Could authors try to analyze another control protein such as tubulin or show red ponceau coloration?
It is important because all changes between groups depends on this. In fact, how the author explains the change in AMPKalpha (total)? It is normally stable.
â–¶ Thanks for the reviewer’s informative recommendation. The tissue was used up in the previous experiment, so we could not conduct the additional western blot experiment with liver tissue. Since the Fig. 5 protein expression results were derived by dividing the β-actin expression level, it is judged to be a significant result. In addition, many studies were explained that AMPK activation (phosphorylation) closely related de novo lipogenesis and fatty acid oxidation in liver, so we focused on determining the AMPK activation level through the ρ-AMPK/AMPK ratio. In future studies, we will compare the blotting results by checking the expression of other control proteins.
Most of the findings suggest that effects of CBA 7346 is linked to the gut-liver axis and then to LPS release in the serum. What is about the histology of the intestine? Authors should give information about this. For example, with goblet cells labeling.
â–¶ We would like to thank the reviewer for pointing out the lack of manuscript. Although the possibility of the gut-liver axis link was confirmed through the LPS release result, the histology analysis of the intestine was not performed because the focus was on only fatty liver during the study.
I have a concern about results of glycemia-insulinemia-HOMA-IR. The authors state that the treatment improves insulin resistance. It is not specifically measured (only HOMA). The blood glucose and insulinemia values ​​must be detailed.
An ITT metabolic test should be performed. It could also be supplemented by a specific pyruvate tolerance test for hepatic metabolism.
Finally, a good way to estimate liver tissue insulin resistance is to do phospho-PKB blots after insulin stimulation. It could be done and interesting things could be observed.
Despite these comments, the work is interesting.
â–¶ Thank for the reviewer’s critical point. It would have been nice to do the ITT experiment together, but it was not performed in consideration of the mouse’s stress. According to the reviewer’s comment, we revised the manuscript in method as below:
Lines 272–274: An alternative to glucose clamp, HOMA-IR is the most commonly used surrogate measure of the insulin resistance in animal experiments [16].
â–¶ In addition, the tissue was used up in the previous experiment, so we could not conduct the additional western blot experiment with tissue. With reference to the reviewer’s comments, we will reflect this so that we can draw conclusions based on various results in future insulin resistance experiments.
※ We would like to thank the reviewers for the constructive and insightful comments and hope that our answers will be acceptable to the reviewers.

Reviewer 3 Report
A good paper that clearly explains the aim, the methods, and the results. I have appreciated the analysis of gut microbiota composition from feces of mice highlighting that the administration of B. uniformis did not significantly influence the microbial composition.
The conclusion supports the results obtained.
Author Response
Reviewer #3:
A good paper that clearly explains the aim, the methods, and the results. I have appreciated the analysis of gut microbiota composition from feces of mice highlighting that the administration of B. uniformis did not significantly influence the microbial composition.
The conclusion supports the results obtained.
â–¶ We appreciate reviewer’s comments about our manuscript. As mentioned by the reviewer, the administration of B. uniformis did not significantly influence the microbial composition in the results of our study, so it was explained that further study is needed in the discussion. Thank you for the positive comments on the manuscript.

Round 2
Reviewer 1 Report
-
Reviewer 2 Report
Thanks to the authors for the quick response effort. The n is ultimately lower than expected for certains experience. I will appreciate the glycemic monitoring (the blood sugar values). Also, even if the food intake is not statistically different, it seems lower in the treated group. This could create a slight interpretation bias. Nevertheless, I find the article interesting.